# Photocatalytic Hydrogen Production and Tetracycline Degradation Using ZnIn_2_S_4_ Quantum Dots Modified g-C_3_N_4_ Composites

**DOI:** 10.3390/nano13020305

**Published:** 2023-01-11

**Authors:** Jingjing Zhang, Xinyue Gu, Yue Zhao, Kai Zhang, Ya Yan, Kezhen Qi

**Affiliations:** 1College of Pharmacy, Dali University, Dali 671000, China; 2Key Laboratory of Advanced Energy Materials Chemistry (Ministry of Education), Nankai University, Tianjin 300071, China; 3College of Biochemistry and Environmental Engineering, Baoding University, Baoding 071000, China; 4College of Science and Technology, Hebei Agricultural University, Cangzhou 061100, China

**Keywords:** ZnIn_2_S_4_ quantum dots, g-C_3_N_4_, photocatalysis, tetracycline degradation, hydrogen production

## Abstract

In this work, ZnIn_2_S_4_/g-C_3_N_4_ (ZIS/CN) composites were synthesized by in-situ growth method, which showed excellent photocatalytic activity in the degradation of tetracycline and hydrogen production from water under visible light irradiation. ZnIn_2_S_4_ quantum dots (ZIS QDs) tightly combined with sheet g-C_3_N_4_ (CN) to accelerate the separation and transportation of photogenerated charges for enhanced photocatalytic activity. Among the prepared nanocomposites, 20%ZnIn_2_S_4_ QDs/g-C_3_N_4_ (20%ZIS/CN) delivered the highest photocatalytic activity. After 120 min of irradiation, the degradation rate of tetracycline with 20%ZIS/CN was 54.82%, 3.1 times that of CN while the rate of hydrogen production was 75.2 μmol·g^−1^·h^−1^. According to the optical and electrochemical characterization analysis, it was concluded that the excellent photocatalytic activities of the composite materials were mainly due to the following three points: enhancement in light absorption capacity, acceleration in the charge transport, and reduction in the carrier recombination rate through the formation of S-scheme heterojunction in the composite system. The high photocatalytic activity of ZIS/CN composites provides a new idea to develop highly efficient photocatalysts.

## 1. Introduction

As a result of the wanton exploitation and use of fossil fuels, environmental pollution and energy shortage are becoming serious issues. As a new low-carbon, clean and efficient technology, hydrogen energy is known as the ultimate energy source in the 21st century [1]. In 1972, Honda and Fujishima first presented the application of TiO_2_ in water to produce hydrogen. Since then, research on photocatalytic hydrogen production of semiconductor materials has risen [2]. Other photocatalysts such as ZnO, CdS and other materials have also been used but their photocatalytic activities are low as they are only sensitive to ultraviolet light and easy to corrode [3,4,5,6,7]. Since visible light accounts for 43% of the total solar spectrum while UV light only accounts for about 4% [8]. Therefore, semiconductor materials applicable at wide range of solar light still need to be explored for photocatalytic hydrogen production from water splitting.

The performance of a photocatalyst mainly depends on three factors: the light absorption capacity, the charge transfer rate, and the surface-active sites for photocatalytic reactions [9]. In 2009, non-metallic polymer carbon nitride (CN) with visible light response appeared unexpectedly. Wang et al. first reported the characteristics of CN for hydrogen production from water [10]. Since then, it has set off a research boom in the field of photocatalysis [11,12,13,14,15]. In the research of photocatalytic hydrogen production, the conduction band (−1.4 eV) and valence band (1.3 eV) potentials of CN meet the thermodynamic conditions (3.0 > Eg > 1.23 eV) for water splitting to generate H_2_ and O_2_ [16]. The disadvantage is that the specific surface area of CN is small, its band gap is narrow, and its catalytic activities are not ideal due to the fast recombination of electron-hole pairs [17]. Therefore, reasonable modification of CN is a necessary means to improve its photocatalytic activity. Methods such as element doping, supporting cocatalysts, and constructing heterojunctions are commonly used material modification strategies [18,19,20,21]. Among them, the heterojunction of narrow band gap semiconductor with an effective means to improve the catalytic activity of CN [22,23,24,25,26,27]. The intimate interface contact between the two materials is conducive to improve the shortcomings of CN such as slow charge separation rate and narrow visible light absorption range [28]. The reported CN composite materials, such as Cd_0.5_Zn_0.5_S/g-C_3_N_4_, Cu_2_MoS_4_/g-C_3_N_4_, CuInS_2_/g-C_3_N_4_, etc., have shown high efficiencies in water splitting to produce hydrogen under visible light irradiation [29,30,31].

In this work, ZIS/CN composites are prepared by in-situ growth using one pot solvothermal method as the synthesis strategy. Under visible light irradiation, the photocatalytic activities of the prepared samples were tested by degrading tetracycline and splitting water to produce hydrogen. Compared with CN, ZIS/CN showed significantly enhanced photocatalytic activities attributed to the more visible light absorption, high charge transfer rate, and high percentage of available active sites for the adsorption and redox reactions during photocatalysis. This work provides a new reference to design and prepare CN-based composite photocatalysts based on “zero dimensional” materials.

## 2. Experimental Section

### 2.1. Material Preparation

The CN nanosheets were prepared by high-temperature calcination method. About 10 g urea was taken in an alumina crucible with a lid and heated in a muffle furnace at 520 °C for 4 h at the heating rate of 10 °C/min. After the completion of the process, the collected light-yellow powder of CN was grinded finely and stored for further experimental work.

The ZIS QDs were prepared by simple hydrothermal method. Accurately weighed amount of Zn(NO_3_)_2_·6H_2_O (0.1 mmol), In(NO_3_)_3_·xH_2_O (0.8 mmol) and GSH were (2 mmol) were dissolved in 40 mL deionized water as solvent under vigorous sonication for 10 min. An amount of 0.05 mol/L NaOH solution was added dropwise to the above solution under magnetic stirring to adjust the pH to 8.5. The basic mixture was then transferred to a three-necked flask connected to an oil bath equipped with a condensation reflux device. After heating at 90 °C under nitrogen protection and magnetic stirring, 2 mmol NaS_2_ solution was added quickly, and heating was continued for 1 h under constant temperature reaction. After the solution was naturally cooled to room temperature, the precipitate was collected by centrifugation for 6 min and washed with isopropanol and deionized water to remove the impurities. The purified precipitate was then dried in vacuum at room temperature. The final ZnIn_2_S_4_ quantum dots product was milky white powder and denoted as ZIS QDs.

Nanocomposites containing different amounts of both CN and ZIS QDs were prepared. Calculated amounts of both CN and ZIS QDs were mixed in ethanol/water mixture under stirring and heated at 80 °C to evaporate the solvent under stirring. The mixture was then dried in an oven at room temperature to obtain the final products. The obtained composite materials were represented by xZIS/CN where ZIS stands for ZnIn_2_S_4_ QDs, CN for g-C_3_N_4_, and “x” for the percent amount (5, 10, 15, 20, 25 wt.%) of ZIS QDs in the nanocomposites.

### 2.2. Characterization

The crystal structure of the samples was analyzed with X-ray powder diffraction (XRD). Transmission electron microscopy (TEM) and high-resolution transmission electron microscopy (HRTEM) were used to observe the micro morphology and structure of the samples. X-ray photoelectron spectroscopy (XPS) was used to analyze the constituent elements and their valence states in the samples. The chemical structure of the sample was characterized by Fourier transform infrared spectroscopy (FTIR). The light absorption properties of the samples were analyzed by UV-Visible absorption spectroscopy using TU-1901 ultraviolet visible spectrophotometer. The photoluminescence (PL) spectra of the sample were measured by RF-5301PC fluorescence spectrometer at the excitation wavelength of 325 nm. FLS1000 fluorescence lifetime spectrophotometer was used to measure the time-resolved PL (TRPL) spectra of the prepared samples.

### 2.3. Photocatalytic Activity

The photocatalytic degradation tests of tetracycline were carried out in self-made photocatalytic reaction device in the laboratory with 1000 W halogen lamp as the light source with a λ > 420 nm cut-on wavelength. About 50 mg photocatalyst was dispersed in 100 mL (40 mg/L) solution of tetracycline and vigorously stirred in dark for some time to obtain adsorption-desorption equilibrium. The sample cup was placed 10 cm away from the light source and the photocatalytic degradation tests were carried out for 120 min under visible light irradiation. About 5 mL solution was sampled after every 20 min, centrifuged, and filtered to remove the solid photocatalyst. Absorbance of the solution was checked with TU-1901 ultraviolet visible spectrophotometer, and the degradation rate of TC was calculated. The molecule structure of tetracycline is as shown in Figure 1. The ultraviolet absorption of TC occurs at 250–270 nm and 343–366 nm, and the absorbance value at 350–366 nm of the maximum absorption peak was taken as the reference [32].

Photocatalytic water splitting for hydrogen production was tested in a photocatalytic device system equipped with Shimadzu GC-14C gas chromatograph. The light source was 350 W Xenon lamp (Beijing NBET Co. Ltd., Beijing, China) with 365 nm filter while the working voltage was 667 V at the current of 13.6 mA. The reaction vessel was a 100 mL three-necked flask, and the port was sealed with a silicone rubber diaphragm. About 20 mg solid photocatalyst was dispersed in a mixture of 90 mL deionized water and 10 mL triethanolamine (TEOA) as the sacrificial agent to quench the photogenerated holes. Before light illumination for the production of hydrogen gas, oxygen gas was evacuated from the solution by bubbling highly pure nitrogen gas for 30 min.

### 2.4. Radical/Charge Capturing Experiment

In order to understand the photocatalytic mechanism of the composite, free radical trapping experiments were carried out. During the photocatalytic degradation processes, 1 mL each of benzoquinone solution (BQ, 0.01 mol/L), n-butanol (n-BA), disodium ethylenediaminetetraacetate solution (EDTA-2Na, 0.01 mol/L) and potassium persulfate (KPS, 0.01 mol/L) was added to capture superoxide radical (^•^O^2−^), hydroxyl radical (^•^OH), hole (h^+^) and electron (e^−^), respectively, before turning on the light. By measuring absorbance of the solution before and after 120 min of illumination, the change in the concentration of TC was determined to find the main active degrading species.

### 2.5. Photoelectrochemical Test

The photoelectric properties including Mott Schottky curve, transient photocurrent (I-t), and impedance spectroscopy (EIS) of CN, ZIS QDs, and 20%ZIS/CN samples were tested by electrochemical workstation (Shanghai Chenhua, CHI660D). The standard three electrodes system was used in the experiment with the saturated calomel electrode (SCE) as the reference electrode, Pt electrode as the counter electrode, and the ITO conductive glass coated with samples as the working electrode in the presence of 0.1 M Na_2_SO_4_ solution as the electrolyte keeping the scanning rate as 10 mV/s, potential scanning range as −0.7~1 V, and scanning frequency range as 0.1 Hz~10 kHz.

Preparation of working electrode: 16 mg of the sample were mixed with 3.6 mL absolute ethanol and 0.4 mL of 5% Nafion solution under stirring for 4 h followed by ultrasonication for 30 min to form a homogenous paste. About 30 μL suspension (4 mg/mL) was applied to the conductive surface of ITO glass having a sample area of about 10 mm × 10 mm. The prepared photocatalysts were allowed to stand for 1 h followed by drying under infrared lamp to obtain the working electrode.

## 3. Results

### 3.1. XRD

The phase structure of the prepared photocatalysts was determined by XRD analysis and the obtained results are presented in Figure 2. CN possesses two characteristic diffraction peaks positioned at 12.9° and 27.6°. These peaks are respectively attributed to the (100) and (002) crystal planes of g-C_3_N_4_ (JCPDS No. 87-1526) and are typically found in two-dimensional graphene-like materials. The former peak at 12.9° is attributed to the stacked triazine structural units while the later peak at 27.6° is assigned to the interlayer stacking of the conjugated aromatic system [33]. For ZIS QDs, the characteristic diffraction peaks at 27.9° and 48.0° are corresponded to the (311) and (440) cubic crystal planes of ZnIn_2_S_4_ (JCPDS No. 72-0305) [34]. In the XRD patterns of ZIS/CN composite, the characteristic diffraction peaks of ZIS QDs at 27.9° and CN at 27.6° overlap due to too close proximity. Also, as the amount of ZIS QDs increases, the diffraction peak of CN at 12.9° gradually disappears with increase in the content of ZIS QDs. The above XRD result shows that ZIS QDs and CN formed heterojunction in the prepared nanocomposite, which promotes the charge separation and transfer to boost the photocatalytic activities.

### 3.2. FTIR

The presence of different functional groups was determined by Fourier transform infrared spectroscopy as shown in Figure 3. The characteristic absorption peaks of CN appear in three regions. The absorption peak at 805 cm^−1^ is caused by the symmetric stretching vibration of triazine ring structure. The 1240, 1316, 1401, and 1633 cm^−1^ absorption peaks are attributed to the stretching vibration of C-N heterocycles and C=N double bonds [35]. The strong absorption peak between 3000–3500 cm^−1^ is attributed to the stretching vibration caused by -NH_2_ and adsorbed H_2_O [36]. The absorption peak at 1610 cm^−1^ is associated with O-H stretching vibration, indicating that water molecules are adsorbed on the sample surface. The Zn-S, In-S, and other chemical bonds in ZIS QDs are difficult to identify in the FTIR spectrum, because the chemical bonds of inorganic substances are extremely weak in FTIR. The absorption band at 1600–1640 cm^−1^ is attributed to the O-H bending vibration peak of water molecules and hydroxyl groups adsorbed on the sample surface [37]. The absorption peak of the ZIS/CN composite is highly similar to the FT-IR spectrum of CN indicating that the composite retains the layered structure of CN. With the increase in the content of ZIS QDs, the strength of the characteristic absorption peak of the composite is gradually weaken indicating that the functional groups in the surface layer of CN are partially consumed. Accordingly, a new chemical bond is formed between CN and ZIS QDs.

### 3.3. TEM

The microstructure and morphology of 20%ZIS/CN composite were characterized by TEM and HRTEM. Figure 4a shows the TEM image of the composite. CN shows a folded lamellar structure, and ZIS QDs grow in situ on its surface. A stable conductive interface contact is formed between ZIS and CN to separate the photogenerated carriers for enhanced photocatalytic performance. Figure 4b is the HRTEM image of 20%ZIS/CN. There are two kinds of lattice stripes, 0.222 and 0.185 nm, which are, respectively, consistent with the lattice spacing of CN (111) and ZIS (440) crystal planes. The size of ZIS QDs lies in the range of 4–6 nm. In addition, the samples were scanned by EDS. As shown in Figure 4c, the five elements C, N, Zn, In, and S are evenly distributed. These results indicate that ZIS QDs are successfully grown on the surface of CN.

### 3.4. XPS

XPS was used to analyze the elemental composition and chemical state of the constituent particles in 20%ZIS/CN composite. Figure 5a shows the full spectrum of 20%ZIS/CN composite indicating that the sample mainly contains C, N, Zn, In, and S and these results are consistent with the EDS results. The binding energy peak at 532.2 eV is attributed to O 1s due to the adsorbed H_2_O on the sample surface. Figure 5b shows the high-resolution spectrum of Zn. The peaks at the binding energies of 1044.7 and 1021.8 eV come from the spin orbit splitting coupling effect of Zn^2+^ corresponding to Zn 2p_1/2_ and Zn 2p_3/2_, respectively [38]. Figure 5c shows that the peaks at 452.2 and 444.7 eV belong to In 3d_3/2_ and In 3d_5/2_, respectively, indicating the existence of In^3+^ state [39]. The binding energies at 168.5, 162.9, and 161.6 eV in Figure 5d are credited to S 2p_3/2_, S 2p_3/2_ and S 2p_1/2_, respectively, indicating that S exists in the form of S^2−^. Figure 5e shows the high-resolution spectrum of C. There are two characteristic peaks at the binding energies of 288.3 and 285.0 eV which are attributed to sp^2^ hybrid carbon (N-C=N bond) and standard graphite carbon, respectively [40]. There is a weak absorption peak at 287.2 eV which can be attributed to the impurity carbon on the sample surface. Figure 5f shows the high-resolution spectrum of N; after peak separation and fitting by XPS peak software, there are three more characteristic peaks at 401.0, 400.0, and 398.8 eV. The main peak at 398.8 eV is the bond energy generated by the participation of N atom in the carbon nitride C=N-C structure of graphite due to sp^2^ hybridization, the peak at 400.0 eV is the bond energy of N-(C)_3_ functional group, while the peak at 401.0 eV is the N-H bond energy due to some residual amino groups that do not participate in thermal polymerization [41].

### 3.5. Photocatalytic Activity

To explore the photocatalytic activity of the as-prepared samples, 100 (40 mg/L) mL TC solution was mixed with the prepared photocatalyst and irradiated under visible light. Figure 6a shows the TC photodegradation curve of CN and ZIS/CN composites. The degradation rate of TC with CN is 17.72% after 120 min exposure to visible light. The photocatalytic degradation performances of different proportions of ZIS/CN are better than that of CN. Obviously, catalytic activity of the materials increases with an increase in the loading proportion of ZIS QDs. Among the nanocomposites, the composite loaded with 20%ZIS QDs has the best degradation ability, and the degradation rate reaches 54.82% in 120 min which is 3.1 times higher than that of CN. When the loading amount of ZIS QDs is continuously increased, the catalytic activity of 25%ZIS/CN decreases slightly, indicating that the shadow effect caused by excessive load is not transparent to the light photons to generate excited electron-hole pairs which weakens the redox ability of the photocatalyst for pollutant degradation. Figure 6b shows the quasi first-order kinetic curve for photocatalytic degradation of TC by CN and ZIS/CN composites. The dynamic constants of ZIS/CN composite material are higher than those of CN. Among them, the apparent rate constant of 20%ZIS/CN (k = 0.00636 min^−1^) is the highest and 4.2 times higher than that of CN.

The ZIS/CN composite also shows enhanced hydrogen generation performance, and its hydrogen generation rate is higher than that of CN and ZIS QDs samples. The hydrogen generation rate for the CN photocatalyst is almost zero. After coupling with ZIS QDs, the hydrogen generation rate is accelerated significantly. Also, as the amount of ZIS QDs increases, the hydrogen production rate is also gradually increased. When the loading amount of ZIS QDs is 20%, the amount of hydrogen generated reaches 75.2 μmol g^−1^ in 1 h. However, further increases in the loading amount of ZIS QDs decreases the hydrogen production as can be seen in case of 25%ZIS/CN composite which delivers 48.5 μmol g^−1^ hydrogen in 1 h (Figure 6c). This may be due to the recombination of photogenerated holes and electrons in the ZIS/CN composite caused by excessive loading of ZIS QDs to retard the hydrogen production rate. Thus, proper amount of ZIS QDs can effectively improve the photocatalytic ability of CN. Figure 6d shows that the hydrogen production rate of 20%ZIS/CN does not reduce obviously in five consecutive cycles indicating that the photocatalyst owns good stability.

### 3.6. Energy Band Structure

Figure 7a shows the UV vis absorption spectra of CN, ZIS QDs and ZIS/CN composites. The light absorption edges of CN and ZIS QDs lie at 460 and 700 nm, respectively. Compared with CN, the light absorption of ZIS/CN composite material has a red shift, and the light absorption capacity increases with increase in the amount of ZIS QDs as from the absorption edge of 20%ZIS/CN at 500 nm.

The Kubelka Munk [42] function was obtained by converting UV-vis data with Tauc formula (Equation (1)). Figure 7b shows the band gap (Eg) diagram of CN, ZIS QDs and 20%ZIS/CN photocatalyst.
(1)$$\alpha h\nu =A{\left(h\nu -E_{g}\right)}^{\frac{n}{2}}$$

Here, α is the light absorption coefficient, h is the Planck constant, ν is the frequency of light, A is the scale constant, Eg is the band gap width of the semiconductor material. and N is a constant. The intercept between the tangent part of Kubelka Munk curve and the abscissa gives the Eg value of the sample which is 3.06 eV for CN, 2.94 eV for ZIS QDs, and 2.98 eV for 20%ZIS/CN. Compared with CN, the Eg value of 2%ZIS/CN composite decreases by 0.08 eV which indicates that the transition resistance of excited electrons is relatively reduced to enhance its photocatalytic ability.

The flat band potential of the sample was measured by impedance potential method. Figure 7c shows the Mott Schottky curve of CN, ZIS QDs, and 20%ZIS/CN. The tangent slopes of the three samples are positive, indicating that the samples are n-type semiconductors. According to the intercept from each tangent line to the x-axis, the flat band potential of CN, ZIS QDs, and 20%ZIS/CN is −0.53 V (vs. SCE), −0.74 V (vs. SCE), and −0.68 V (vs. SCE), respectively. In n-type semiconductors, the conduction band potential is approximately equal to the flat band potential [43]. The valence band potential (E_VB_) of the sample was calculated according to Equation (2) which is 2.53 eV for CN), 2.2 eV for ZIS QDs, and 2.3 eV for 20%ZIS/CN.
(2)$$\;E_{VB}=E_{CB}+E_{g}$$

In Equation (2) *E_VB_*, *E_CB_*, and *E_g_* are the valence band potential, conduction band potential, and band gap, respectively.

Figure 7d shows the energy band structure of CN, ZIS QDs and 20%ZIS/CN. The Eg value of the composite material is smaller by 0.08 V than that of CN, indicating that the composite material can generate more photogenerated carriers under long wavelengths and has a stronger ability to use visible light.

### 3.7. Photoelectric Performance

The separation, transport, and recombination rate of photogenerated electrons and holes is another important index to evaluate the photocatalytic activity of materials. In this work, transient photocurrent response experiment, electrochemical impedance spectroscopy (EIS), and fluorescence spectroscopy (PL) were used to analyze the generation and transportation of charges in CN, ZIS QDs and 20%ZIS/CN nanocomposite.

Photocurrent is a characterization technique to study the photogenerated electron mobility in photocatalytic materials. In general, the stronger photocurrent, the higher the separation efficiency of electrons and holes, and better the photocatalytic activity [44]. Figure 8a shows the photocurrent spectrum of CN, ZIS QDs, and 20%ZIS/CN nanocomposite. The sample has a stable and repeatable photocurrent response. The order of photocurrent intensity is as follows: ZIS QDs > 20%ZIS/CN >CN. Photocurrent density of 20%ZIS/CN (0.1820 μA/cm^2^) is higher than CN (0.0773 μA/cm^2^) indicating that the introduction of ZIS QDs promotes the carrier separation ability of CN. Thus, more free charges are excited and transferred to the surface of the photocatalyst for improved photocatalysis.

The electrochemical impedance spectroscopy is a useful method to characterize the ability of charge transfer, which is an important factor to determine the efficiency of catalysts. Figure 8b shows the EIS spectra of CN, ZIS QDs, and 20%ZIS/CN. The semicircle arc diameter represents the charge transfer resistance (Rct). The smaller the radius of the EIS semicircle, the smaller the resistance, and consequently the higher the charge transfer efficiency. Compared with CN, 20%ZIS/CN has smaller interface impedance and better charge transfer capability. The ZIS QDs half arc is the smallest indicating excellent conductivity, which is highly consistent with the strong photocurrent on the I-t curve.

The photoluminescence study provides information about charge recombination in the material. Generally, the stronger the emission peak intensity, the higher the electron hole recombination rate in the catalyst. Figure 8c shows the steady-state PL spectra of CN, ZIS QDs, and ZIS/CN composites. Compared with a CN, the PL intensity of ZIS/CN is reduced which indicates that the introduction of ZIS QDs decreases the recombination of photogenerated electrons and holes in the composite photocatalyst. The time-resolved PL spectrum further supports the enhanced charge separation rate through formation of the heterojunction between CN and ZIS QDs. As shown in Figure 8d, the average PL lifetime of 20%ZIS/CN is 3.77 ns which is shorter than that of CN as 4.90 ns. This result confirms that more effective separation of photogenerated electron-hole pairs in ZIS/CN for enhanced photoactivity. It can be concluded that the introduction of an appropriate amount of ZIS QDs over CN surface can significantly enhance the charge separation and transfer ability while reducing the carrier recombination rate for excellent catalytic activity.

### 3.8. Photocatalytic Mechanism

Determination of free radicals during the oxidation of pollutants provides sufficient information about the degradation mechanism. In this work, 20%ZIS/CN photocatalyst was taken as an example to conduct active species capture experiment and the results are shown in Figure 9. The degradation rate of TC without capture agent is 54.82%, and repeated experiments show that the sample has good stability. The degradation rates of TC in the presence and absence of capturing agents such as BQ, n-BA, EDTA-2Na, and KPS after 120 min of visible light irradiation are in the order of None > KPS> n-BA >EDTA-2Na > BQ. These results show that e^−^, h^+^, ^•^OH and ^•^O^2−^ all participated in the photocatalytic degradation of TC over 20%ZIS/CN, and the main active species were ^•^O^2 −^ > h^+^ > ^•^OH >e^−^.

Based on the obtained results and discussion, a detailed photocatalytic mechanism of the photoexcited charge transfer in ZIS QDs/CN composite is depicted in Figure 10. It can be observed that when ZIS and CN are in contact, the electrons in ZIS spontaneously transfer via the interface to CN until both attain the same energy level. Remarkably, in this process, ZIS QDs produce excited electrons and possess a positive charge whereas CN gains electrons and carries a negative charge at the interface. Accordingly, an internal electronic field is accomplished at the interface position which is directed from ZIS QDs to CN to significantly facilitate the charge transfer and separation. Meanwhile, the band edge of ZIS QDs is positioned upward due to the loss of electrons, while that of CN bents downward owing to electron receiving ability. Furthermore, under photo irradiation, the electrons of ZIS QDs and CN are accordingly excited from their VBs to the CBs. As a result of the internal electric field, band edge bending, and coulomb interaction environment, the photoexcited electrons with low energy in the conduction band of CN effectively combine with the available positive charged holes in the VB of ZIS QDs, the S-scheme heterojunction is successfully established [45]. As mentioned above, this S-scheme effectively improves photocatalytic activities.

## 4. Conclusions

A series of ZIS QDs/CN heterojunction photocatalysts were prepared by simple in-situ growth method and used to degrade the tetracycline antibiotic in water and produce hydrogen from water splitting. By optimizing the amount of ZIS QDs in the heterogeneous catalyst, the optimized 20%ZIS/CN sample showed 54.82% degradation efficiency in 120 min under visible light irradiation much higher than those of ZIS QDs and CN alone. The optimized sample also showed the highest hydrogen production rate of 75.2 μmol/h compared to CN. These enhanced photocatalytic activities were attributed to the well-established conducive heterojunctional interface between ZIS QDs and CN to separate the photogenerated charges efficiently. The capture experiments showed that hydroxyl radicals were the main active species to degrade pollutants under visible light irradiation. Based on the optical and electrochemical characterization analysis, the separation of excited charges was achieved through S-scheme heterojunction in the nanocomposite. This work will provide more ideas for develop highly efficient photocatalysts for application of environment and energy.

## Figures and Tables

**Figure 1 nanomaterials-13-00305-f001:**
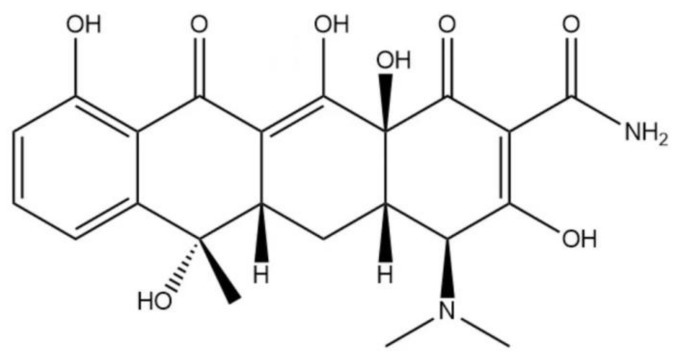
The molecule structure of tetracycline.

**Figure 2 nanomaterials-13-00305-f002:**
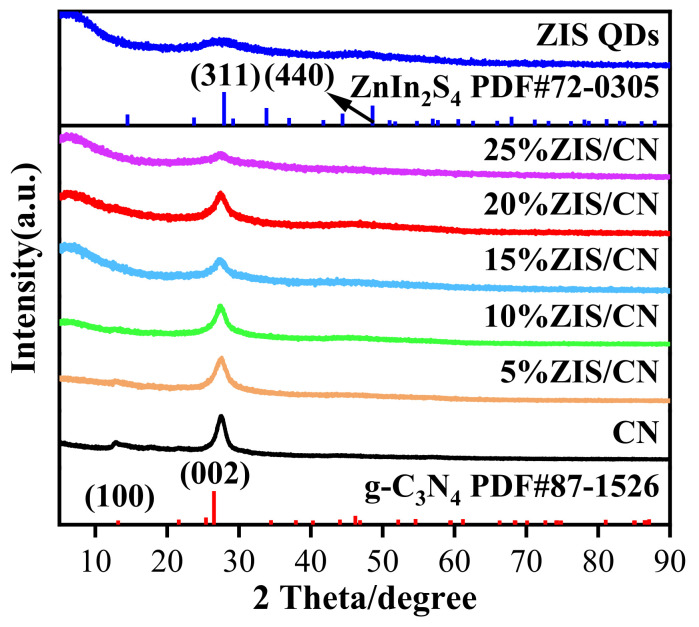
XRD patterns of CN, ZIS QDs and ZIS/CN.

**Figure 3 nanomaterials-13-00305-f003:**
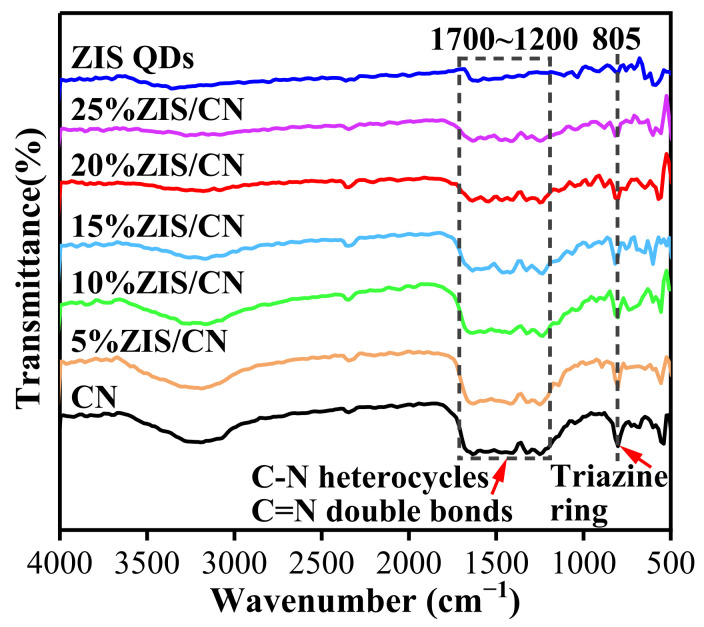
FT-IR Spectra of CN, ZIS QDs, and ZIS/CN.

**Figure 4 nanomaterials-13-00305-f004:**
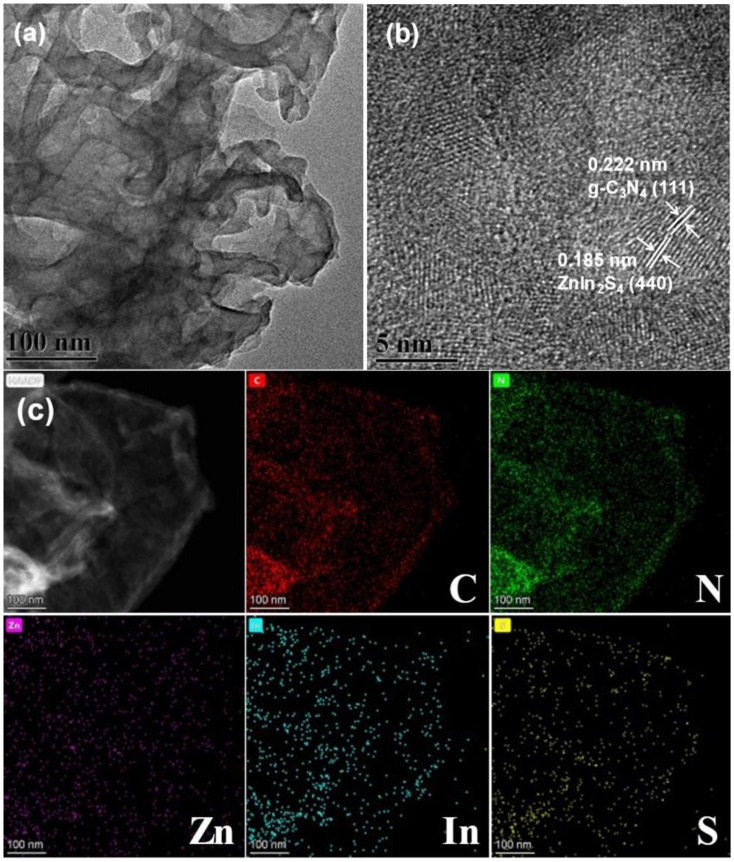
(**a**) TEM (**b**) HRTEM and (**c**) EDS element mapping of 20% ZIS/CN.

**Figure 5 nanomaterials-13-00305-f005:**
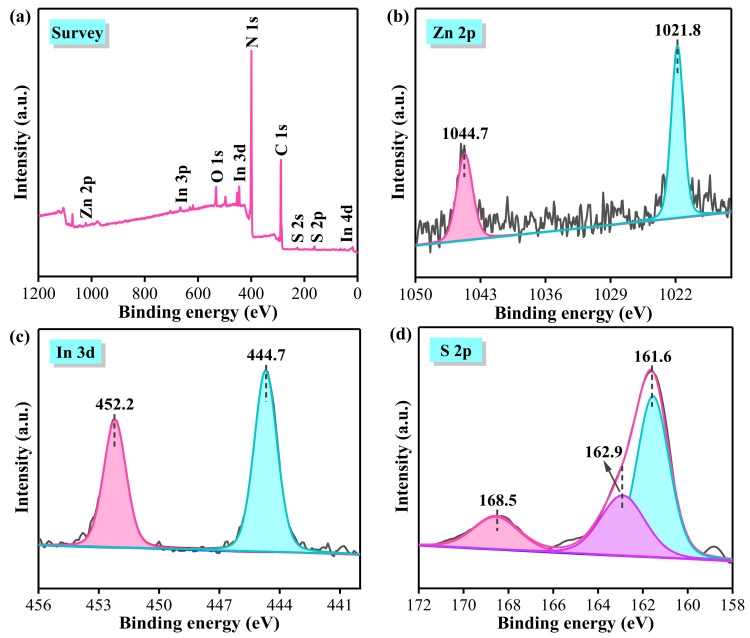
XPS spectrum of 20%ZIS/CN: (**a**) survey XPS spectra; (**b**) Zn 2p; (**c**) In 3d; (**d**) S 2p; (**e**) C 1s; (**f**) N 1s.

**Figure 6 nanomaterials-13-00305-f006:**
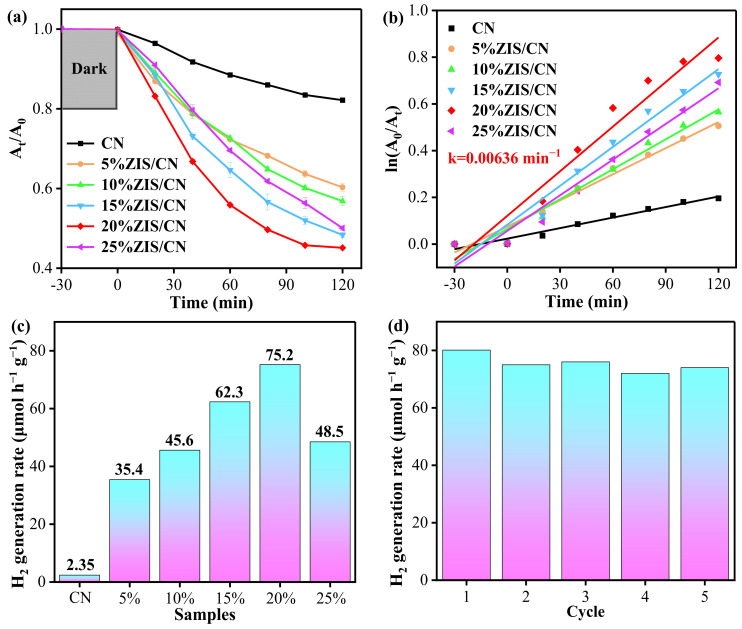
(**a**) TC degradation curve of CN and ZIS/CN composites; (**b**) kinetic curve of TC photodegradation; (**c**) hydrogen production rates; (**d**) recycling test of the photocatalytic hydrogen generation over 20%ZIS/CN.

**Figure 7 nanomaterials-13-00305-f007:**
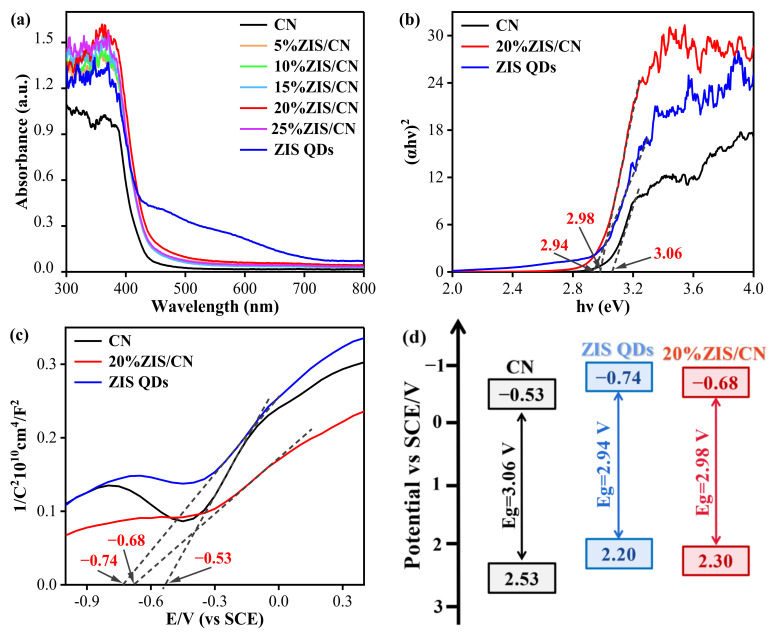
Photoelectric property of CN, ZIS QDs, and 20%ZIS/CN samples: (**a**) UV–Vis DRS spectra, (**b**) energy band gap, (**c**) flat band potential spectrogram, (**d**) energy band structure diagram.

**Figure 8 nanomaterials-13-00305-f008:**
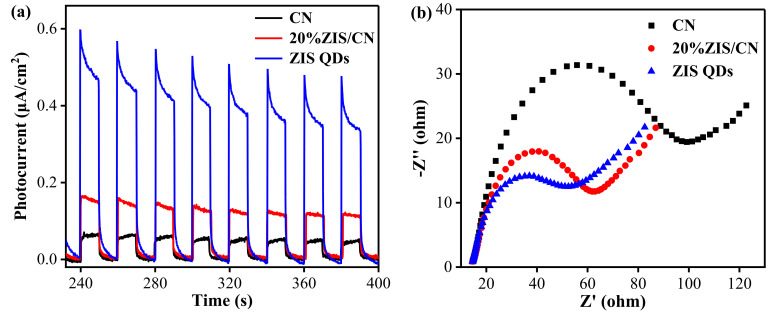
Photoelectric property of CN, ZIS QDs and 20%ZIS/CN samples: (**a**) photocurrent time curve, (**b**) electrochemical impedance spectrum, (**c**) steady-state fluorescence spectra, (**d**) transient fluorescence.

**Figure 9 nanomaterials-13-00305-f009:**
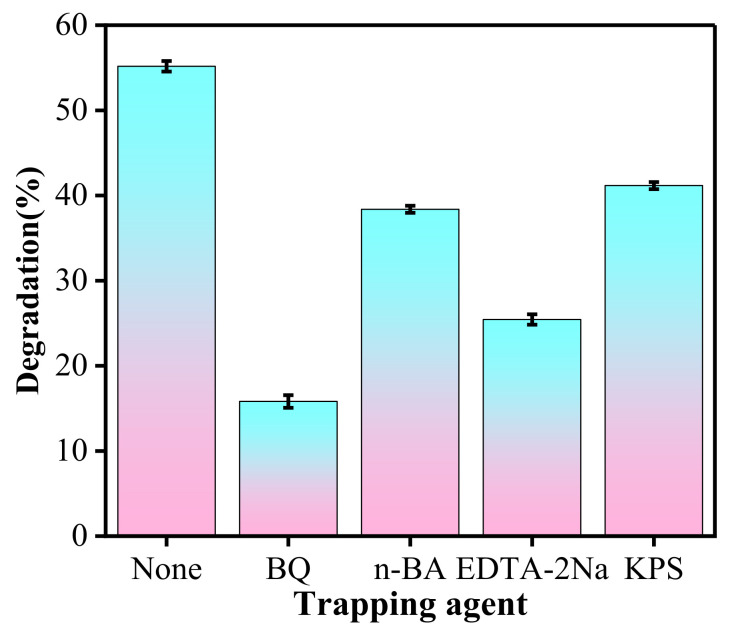
Radical capture experiment of 20%ZIS/CN for photodegradation of TC.

**Figure 10 nanomaterials-13-00305-f010:**
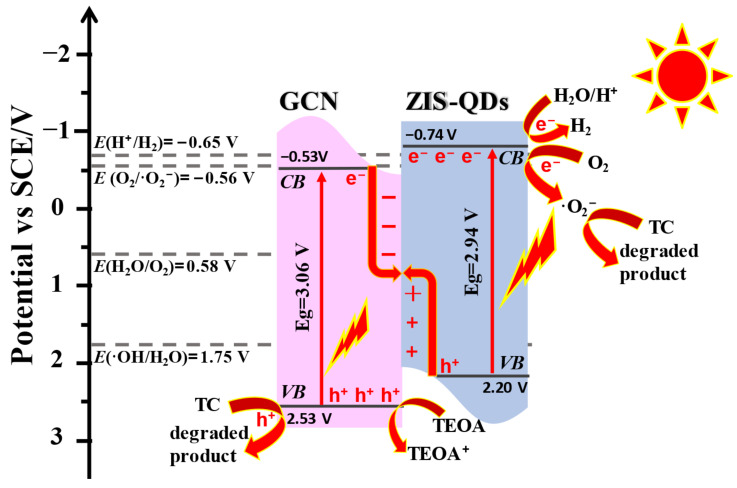
Photocatalytic mechanism of the ZIS/CN composite.

## Data Availability

Not applicable.

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
