# Peer review of "Photocatalytic Hydrogen Production and Tetracycline Degradation Using ZnIn_2_S_4_ Quantum Dots Modified g-C_3_N_4_ Composites"

_nanomaterials, 2023, doi:10.3390/nano13020305_

Round 1

Reviewer 1 Report

line 81 InN3O9·xH2O - In(NO3)3 ?

line 87  "the precipitate was collected by centrifugation" - RCF + time needed

line 164 "With the increase of the proportion of ZIS QDs, this peak gradually becomes wider and shorter indicating that the introduction of ZIS QDs weakens the interlayer stacking force between carbon nitride layers."

Taking into account the overlapping of the peaks and the significant broadening of the ZIS peak, this explanation is doubtful. More likely, it is simply a change in the shape of the peak as a result of a change in the ratio of the components of the composite.

line 167 "The absence of impurity peaks indicates that the prepared nanocomposite is composed of only ZIS QDs and CN"

The statement is not 100% true, there may be amorphous phases.

Fig. 2
How can one explain the noticeable difference (between the 15% sample and the rest) in the 500-600cm-1 region?

"Radical/Charge capturing experiment" is not well described. The result of a decrease in the rate of decomposition may also be related to other factors besides the capture of charge carriers. Has this been taken into account?

Author Response

Please find Details in the attachment.

Reviewer 2 Report

The article "photocatalytic hydrogen production and tetracycline degradation using ZnIn2S4 qds modified g-C3N4 composites is of great interest.

However there are some needs for the article to be published.

1- The readers of the review "Nanomaterials" maybe don not know about the structure of tetracycline. It should be clearly drwn in the manuscript.

2- In your introduction, you present tetracycline as a model pollutant. I disagree. This is not only a neutral molecule you can use as a dye to see the capability of your process, this is a real antibiotic with high side effects, especially on children teeth or pregnant women bones. So the water pollution with this molecule is a real issue, and the introduction (and result/discussion) should take it into consideration.

3- for photocatalytic activity, the authors use a sun-modeling white light source at 1000 watt, but the sample cup was only at 10 cm from this source. Is this accurate compared to sunlight if the process is used in the environment one day?

4- I really appreciate the effort for the photocatalytic mechanism determination. However tetracycline containing phenolic compounds, what if its degradation leaks more toxic adducts? Did the authors perform an analytical assay of their tetracycline degradation to see in mass spec for example the products of this degradation?

With these questions answered, I recommend this article for publication.

Author Response

Please check the details in the attachment.

Round 2

Reviewer 2 Report

Thanks a lot for the corrections / explanations